# Identification of interferon-stimulated genes that attenuate Ebola virus infection

Makoto Kuroda[1], Peter J. Halfmann ●[1 ✉], Lindsay Hill-Batorski[1], Makoto Ozawa ●[2,3], Tiago J. S. Lopes[1], Gabriele Neumann[1], John W. Schoggins ●[4], Charles M. Rice[5] & Yoshihiro Kawaoka ●[1,6,7 ✉]

The West Africa Ebola outbreak was the largest outbreak ever recorded, with over 28,000 reported infections; this devastating epidemic emphasized the need to understand the mechanisms to counteract virus infection. Here, we screen a library of nearly 400 interferon-stimulated genes (ISGs) against a biologically contained Ebola virus and identify several ISGs not previously known to affect Ebola virus infection. Overexpression of the top ten ISGs attenuates virus titers by up to 1000-fold. Mechanistic studies demonstrate that three ISGs interfere with virus entry, six affect viral transcription/replication, and two inhibit virion formation and budding. A comprehensive study of one ISG (CCDC92) that shows anti-Ebola activity in our screen reveals that CCDC92 can inhibit viral transcription and the formation of complete virions via an interaction with the viral protein NP. Our findings provide insights into Ebola virus infection that could be exploited for the development of therapeutics against this virus.

[1] Department of Pathobiological Sciences, School of Veterinary Medicine, Influenza Research Institute, University of Wisconsin–Madison, Madison, WI 53711, USA. [2] Laboratory of Animal Hygiene, Joint Faculty of Veterinary Medicine, Kagoshima University, Kagoshima 890-0065, Japan. [3] Transboundary Animal Diseases Center, Joint Faculty of Veterinary Medicine, Kagoshima University, Kagoshima 890-0065, Japan. [4] Department of Microbiology, UT Southwestern Medical Center, Dallas, TX 75390, USA. [5] Laboratory of Virology and Infectious Disease, Center for the Study of Hepatitis C, The Rockefeller University, New York, NY 10065, USA. [6] Division of Virology, Department of Microbiology and Immunology, International Research Center for Infectious Diseases, Institute of Medical Science, University of Tokyo, Tokyo 113-8654, Japan. [7] ERATO Infection-Induced Host Responses Project, Japan Science and Technology Agency, Saitama 332-0012, Japan. ✉email: peter.halfmann@wisc.edu; yoshihiro.kawaoka@wisc.edu

Ebola virus infection causes a rapid and severe disease in humans that is characterized by coagulation and bleeding abnormalities, an exaggerated inflammatory response, hypotensive shock, and often death[1]. Ebola virus is a member of the *Filoviridae* family, and six virus species in the genus *Ebolavirus* have been identified to date: *Zaire, Sudan, Tai Forest* (previously *Cote d'Ivoire*), *Reston, Bundibugyo,* and *Bombali*[2,3]. Among these six species, *Zaire ebolavirus* (EBOV) causes the highest case fatality rates in humans, and was the species responsible for the 2014–2016 EBOV outbreak in West Africa. That outbreak was the largest on record with more than 28,000 reported infections and over 11,000 deaths[4]. At the time writing, the second largest outbreak is ongoing in the Democratic Republic of Congo[5].

The EBOV genome is a negative-sense RNA genome that encodes at least seven known structural proteins. The EBOV glycoprotein (GP) mediates virus entry[6,7], whereas four structural proteins—nucleoprotein (NP), RNA-dependent RNA polymerase (L), VP30, and VP35—are important for viral genome amplification[8]. EBOV VP40 is a membrane-associated viral protein that is essential for viral budding[9]. EBOV VP24 and VP35 are key components of the nucleocapsid[10], with VP24 facilitating correct nucleocapsid assembly[11].

The type I interferon (IFN) system, which comprises IFNα and IFNβ, is a key component of the innate immune response and is involved in the control of viral infection. IFNα and IFNβ are activated upon EBOV infection; however, their activation can be counteracted by VP35, which inhibits the phosphorylation and subsequent nuclear translocation of interferon regulatory factor 3 (IRF3)[12]. When IFN is released from infected cells, it binds to IFN receptors on neighboring cells, resulting in the activation of JAK/STAT-dependent signaling pathways. The activation of JAK/STAT pathways can be counteracted by VP24 through the inhibition of STAT-1 nuclear translocation[13]. Activation of JAK/STAT pathways leads to the induction of several hundred interferon-stimulated genes (ISGs)[14]. The ISG products may directly limit viral replication[14–16]; however, once again, EBOV has evolved countermeasures against the action of ISGs; for example, VP35 blocks PKR activation and GP blocks BST2/tetherin-mediated restriction of viral budding[17,18]. This rapid and potent attenuation of antiviral IFN responses likely contributes to the overall pathogenicity of EBOV.

Previous large-scale screening studies have identified multiple ISGs with activity against RNA and DNA viruses[15,19–25]. However, such comprehensive studies are lacking for EBOV. Here, we screen a protein expression library of known ISGs against our previously established biologically contained EBOV (which lacks the essential *VP30* gene and can be used in BSL-2 containment[26,27]) and identify several ISGs that were not previously known to interfere with the EBOV life cycle. Our findings provide insights into Ebola virus infection that could be exploited for the development of antivirals to combat this virus.

## Results

**A luciferase-based screen for identifying ISGs with anti-EBOV activity.** To identify ISGs with anti-EBOV properties, we used a biologically contained EBOV (based on the genome sequence of *Zaire ebolavirus*, Mayinga 1976) that lacks the essential *VP30* gene and expresses the Renilla luciferase reporter gene instead (EBOVΔVP30-luc). This reporter virus replicates efficiently in cell lines stably expressing EBOV VP30, such as human embryonic kidney (HEK)-293T VP30 cells[26]. HEK-293T VP30 cells were transiently transfected with individual protein expression vectors from a library of 389 different ISGs[15]. Twenty-four hours later, the transfected cells were infected with EBOVΔVP30-

luc. Three days post infection, the cells were lysed and virus-driven luciferase expression levels were analyzed. All data were normalized to luciferase levels from cells transfected with a control vector expressing a fluorescent protein that does not inhibit infection and are represented in a dot plot as relative luciferase activities (Fig. 1a; raw data in Source Data). Overexpression of most ISGs reduced viral-driven luciferase expression to some extent (averaging 15% inhibition, indicated by the solid vertical line in Fig. 1a). For further evaluation, we selected 21 ISGs that caused a statistically significant reduction in virus-driven luciferase expression levels ($p$-value ≤ 0.05) and one additional ISG, IFI6, ($p$-value = 0.075) that was previously reported to have antiviral properties[15,20,28] (Fig. 1a; indicated by the black dots).

For the selected 22 ISGs, we performed three independent experiments with EBOVΔVP30-luc infection of transfected HEK-293T VP30 cells and confirmed the anti-EBOV nature of all 22 ISGs (Fig. 1b). To further examine these ISGs in functional studies, we cloned them into the mammalian protein expression vector pCAGGS, which contains the chicken β-actin promoter[26]. When expressed under the control of this promoter, several of the ISGs (i.e., IFIH1/MDA5, IRF1, C9orf91, IRF7, CD9, CTCFL, and CX3CL1) exhibited cytotoxic effects and were, therefore, excluded from further testing. Of the remaining ISGs, we selected the 10 ISGs with the strongest attenuating effect on EBOV-driven luciferase expression (listed in Supplementary Table 1) with little to no inhibition of vector-based, cellular expression of firefly luciferase or Renilla luciferase (i.e., independent of virus replication; Supplementary Fig. 1a–c) for further characterization

**Effect of selected ISGs on EBOVΔVP30 titers.** Next, we tested the effect of ISG overexpression on EBOVΔVP30 titers. Overexpression of all 10 ISGs significantly attenuated virus growth on both days 3 and 6 post infection relative to either the empty pCAGGS vector or pCAGGS-GFP controls (Fig. 1c). The three ISGs with the most significant inhibitory effect, PFKFB3, BTN3A3, and CCDC92, decreased EBOVΔVP30 titers by nearly 1000-fold on day 6 post infection (Fig. 1c).

In a concurrent experiment, cell lysates were collected from transfected, but not infected HEK-293T VP30 cells on days 4 and 7 post transfection to examine the protein levels of each ISG and VP30. In most cases, the protein levels of each ISG were highest on day 4 after transfection and the levels of VP30 remained consistent throughout (Supplementary Fig. 2a). In a separate set of experiments, HEK-293T VP30 cells were transfected and infected as described above and cell viability was assessed. Overexpression of the ISGs tested here did not markedly reduce cell viability compared with either the empty pCAGGS vector or pCAGGS-GFP controls (Supplementary Fig. 2b). Collectively, these data suggest that the virus titer reduction induced by the selected ISGs is virus specific and not due to a significant decrease in VP30 expression in the cell line or to overall cytotoxicity.

Given that EBOV infection of endothelial cells is linked to pathogenesis[29], we next assessed the potential of these ten ISGs to inhibit EBOV infection in primary human umbilical vein endothelial cells stably expressing VP30 (HUVEC VP30 cells). Cells were transfected with each ISG or GFP expression vector by electroporation. Transfected cells with greater than 75% cell viability at 24 h post transfection were infected with EBOVΔVP30-luc (Supplementary Fig. 3a). Overexpression of all the ISGs, except for MAP3K5 and IFI6 whose expression was cytotoxic in HUVEC VP30 cells, significantly reduced EBOV-driven luciferase expression at three days post infection (Supplementary Fig. 3b) in a similar manner of inhibition to that observed in HEK-293T VP30 (Fig. 1b). In a parallel

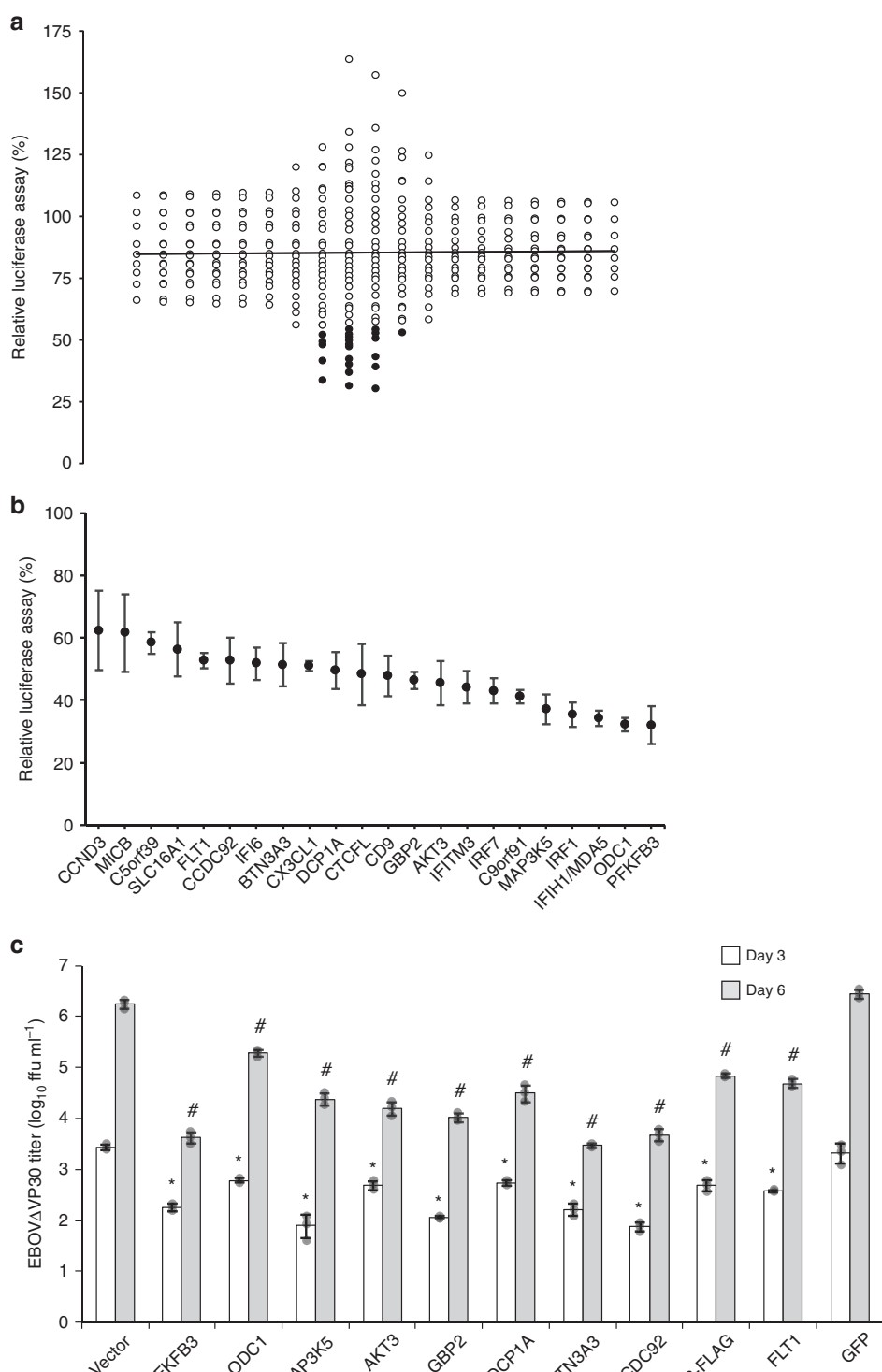

experimental, we confirmed the expression of each ISG in HUVEC VP30 cells after transfection and that the selected ISGs did not decrease the expression of VP30 in the HUVEC VP30 cells by western blot analysis (Supplementary Fig. 3c).

**Anti-EBOV activity of endogenous BTN3A3 induced by IFNγ.** Having demonstrated the antiviral activity of ISGs against EBOV using vectors to overexpress individual ISGs, we next asked whether endogenous expression of any of the top ten ISGs contributed to anti-EBOV activity. Treatment of human cervical

cancer (HeLa) cells with different interferons (IFNα, IFNβ, or IFNγ) for 24 h induced the mRNA expression of many of the ISGs of interest (Supplementary Table 2). We then attempted to confirm protein expression of the ISGs after IFN treatment in HeLa cells by immunoblotting using ISG-specific antibodies. However, we were only able to detect the protein expression of the ISG, BTN3A3, after IFNγ treatment in wild-type HeLa cells (Supplementary Fig. 4) and HeLa cells that stably express VP30 (HeLa VP30 cells; Fig. 2a). When cells were treated with different concentrations of IFNγ for 24 h and then infected with EBOVΔVP30, we observed up to a 100-fold reduction in virus

**Fig. 1 Identification of ISGs that attenuate EBOV-driven luciferase activity and EBOV growth kinetics. a** Dot plot of EBOV-driven luciferase activity in the presence of overexpressed ISGs. HEK-293T VP30 cells were transfected with expression vectors for each ISG for 24 h prior to infection with EBOVΔVP30-luc at an MOI of 1.0. Virus-driven Renilla luciferase activity was measured 48 h post infection and compared with the luciferase activity of infected, control cells that were transfected with a vector expressing only a red fluorescent protein. The black line indicates the ISG population mean, and black filled-in dots represent the top 22 ISGs with significant anti-EBOV activities, which were evaluated further. **b** Confirmation assays with EBOVΔVP30-luc infection of transfected HEK-293T VP30 cells with the top 22 ISGs with anti-EBOV activity from the primary screen. Data are presented as means ± standard deviation (SD), and are representative of three independent experiments. The expression of each of the top 10 ISGs selected for further characterization under this screening condition was confirmed by western blotting (Supplementary Fig. 20). **c** Titers of EBOVΔVP30-GFP from HEK-293T VP30 cells after overexpression of selected ISGs cloned into the protein expression vector pCAGGS. Cells were transfected with expression vectors for each ISG or control vectors for 24 h prior to infection with EBOVΔVP30-GFP at an MOI of 0.0001. Virus titers were then measured on days 3 and 6 post infection. Data are presented as means ± SD, and are representative of three independent experiments. (*) indicates a statistically significant difference (p-value ≤ 0.05) from the day 3 controls; (#) indicates a statistically significant difference (p-value ≤ 0.05) from the day 6 controls. Source data are provided as a Source Data file.

titers on day 2 post infection with no cytotoxicity (Fig. 2b), which is consistent with previous reports that IFNγ inhibits EBOV infection[30,31]. To assess the contribution of endogenous BTN3A3 to IFNγ-mediated inhibition of EBOV, we determined virus titers after IFNγ treatment by reducing *BTN3A3* gene expression via siRNAs targeting two different sites of its mRNA transcript. The efficiency of knockdown was evaluated by use of qRT-PCR; *BTN3A3* mRNA expression increased by 10-fold upon IFNγ treatment and *BTN3A3* mRNA expression was reduced by siRNA treatment (Fig. 2c). In the same experimental setting, HeLa VP30 cells transfected with the siRNAs and then treated with IFNγ were infected with EBOVΔVP30. Virus titers in control siRNA-treated cells were attenuated by 10-fold, whereas virus titers in BTN3A3 siRNA-treated cells were significantly less attenuated (Fig. 2d). In the absence of IFNγ treatment, *BTN3A3* gene knockdown did not affect virus titers (Fig. 2d). We observed similar results when the same experiments were performed in human hepatocarcinoma (Huh7.0) cells (Supplementary Table 2 and Supplementary Fig. 5a–d).

To determine whether stable expression of BTN3A3 could inhibit EBOV growth at endogenous expression levels, we generated HeLa VP30 cells that stably expressed BTN3A3. We observed that BTN3A3 protein expression levels in HeLa cells stably expressing BTN3A3 were similar to those observed in wild-type HeLa VP30 cells treated with IFNγ (Fig. 2e). The titer of EBOVΔVP30 was 50% lower in HeLa VP30 cells stably expressing BTN3A3 compared with that in wild-type HeLa VP30 cells (p = 0.024; Fig. 2f). Taken together, these results indicate that endogenous BTN3A3 has anti-EBOV activity.

**The effect of ISGs on EBOV cell entry**. To gain insight into the mechanism(s) by which ISGs attenuate EBOV infection, we first examined the effect of the selected ISGs on viral entry into host cells. EBOV binding to host cells, virion internalization, and membrane fusion are mediated by GP, which is expressed on the virion surface[32]. To quantitatively assess the effect of each ISG on EBOV GP-mediated viral entry, HEK-293T cells were transfected with pCAGGS vectors expressing individual ISGs, a control vector expressing GFP, or a positive control vector encoding IFITM3, a known ISG that inhibits EBOV and vesicular stomatitis virus (VSV) cell entry[33,34]. Twenty-four hours later, cells were infected with a firefly luciferase-expressing VSV lacking the VSV-*G* gene and pseudotyped with either EBOV GP (VSV-EBOV GP) or VSV-G. Twenty-four hours post infection, cells were lysed and luciferase activity was assessed. As expected, overexpression of the IFITM3 positive control significantly inhibited cell entry of both VSV-EBOV GP and VSV-G, reducing luciferase activity by 85% and 75%, respectively (Fig. 3a). Cell entry of VSV-EBOV GP was also significantly reduced by overexpression of AKT3 (60% reduction), whereas overexpression of AKT3 did not interfere

with cell entry of VSV-G (Fig. 3a). Overexpression of ODC1 and FLT1 significantly reduced cell entry of both VSV-G and VSV-EBOV GP (Fig. 3a). The overexpression of each ISG was confirmed in the lysates of the transfected HEK-293T cells at 24 h post transfection (Supplementary Fig. 6).

Furthermore, we assessed whether these three ISGs had broad antiviral entry activity. Cell entry of VSV pseudotyped with the Lassa virus glycoprotein complex (VSV-LASV GPC) or influenza A/WSN/33 (H1N1) virus hemagglutinin (HA) and neuraminidase (NA) (VSV-WSN HA/NA) was significantly reduced by ODC1 overexpression, whereas no significant inhibition of entry of these viruses was induced by AKT3 or FLT1 (Supplementary Fig. 7a–c).

We then attempted to verify the antiviral activity of ODC1 by using a different entry assay based on a pseudotyped retrovirus. In transfected cells expressing the retrovirus GFP reporter vector pMXs-IRES-GFP (pMXs-IG), we observed no inhibition of GFP expression when OCD1 was co-expressed compared to cells transfected with pMXs-IG and an empty vector (Supplementary Fig. 8a), indicating that OCD1 had no inhibitory effects on the expression of GFP from the retrovirus vector. However, in cells expressing OCD1 and then infected with a VSV glycoprotein pseudotyped retrovirus, we observed a reduction in GFP expression compared to infected cells transfected with an empty vector (Supplementary Fig. 8b). These additional data indicate that the inhibition induced by ODC1 is specific to virus entry mediated by the different viral glycoproteins and not the inhibition of the reporter genes used in the entry assays.

**The effect of ISGs on EBOV transcription/replication**. Next, we tested the effect of ISG overexpression on EBOV transcription/replication in a firefly luciferase-based EBOV minireplicon system. Six ISGs (PFKFB3, MAP3K5, GBP2, DCP1A, CCDC92, and IFI6) significantly reduced viral replication/transcription in the EBOV minireplicon assay (Fig. 3b). ISG overexpression did not significantly reduce cell viability (Supplementary Fig. 9a) or the protein expression level of NP or VP35 (Supplementary Fig. 9b), suggesting that the reduced EBOV transcription/replication activity was specific and not due to ISG-mediated toxicity, general inhibition of cellular processes, or a reduction in viral protein expression. For each ISG, its overexpression was verified by western blot analysis (Supplementary Fig. 9b).

**The effect of ISGs on EBOV virion formation and budding**. The EBOV matrix protein VP40 has a central role in virus assembly and budding[9]. Expression of VP40 alone results in the formation of virus-like particles (VLPs), and co-expression of EBOV GP and NP results in the incorporation of these viral proteins into the VLPs and enhanced budding[35,36]. To determine

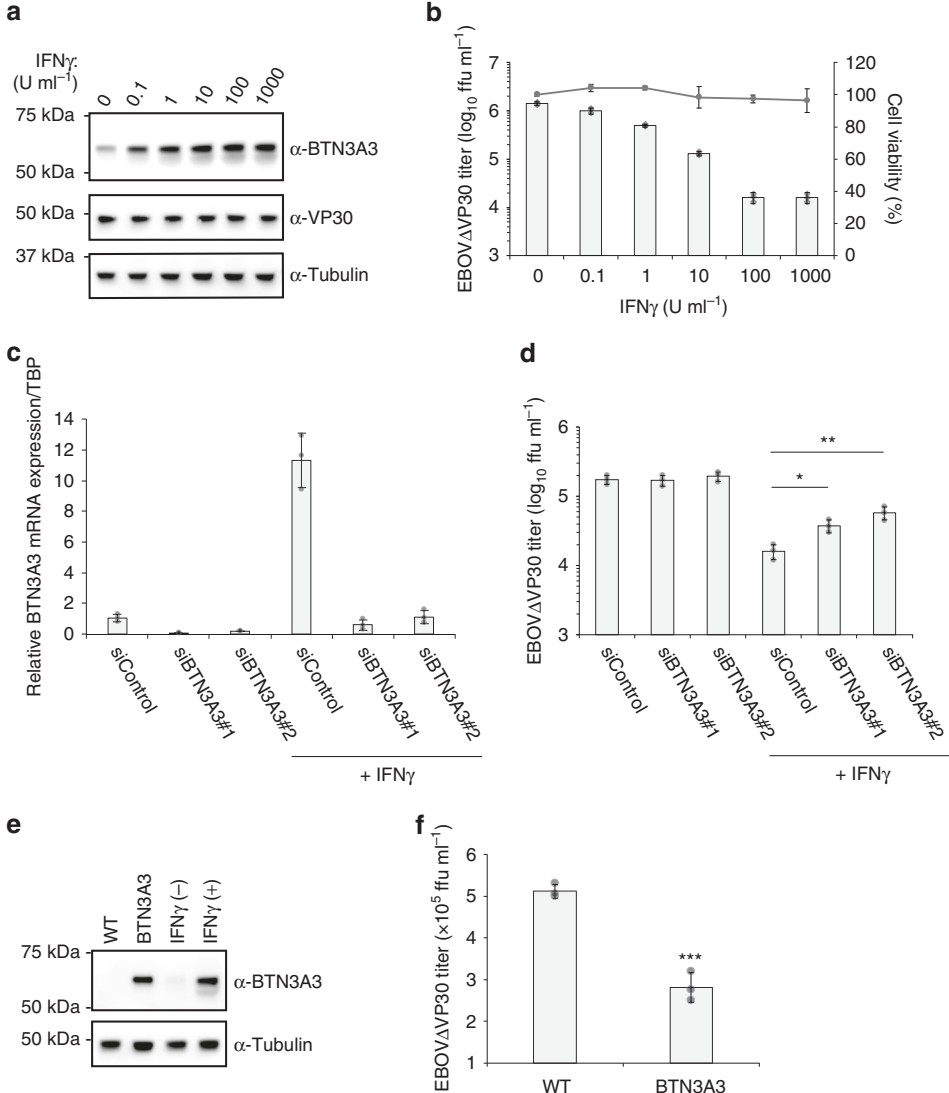

**Fig. 2 EBOV attenuation by endogenous BTN3A3. a** BTN3A3 expression in HeLa VP30 cells treated with the increasing doses of IFNγ for 24 h. Cells were lysed and the indicated protein expression levels were analyzed by immunoblotting. Data are representative of two independent experiments. **b** Titers of EBOVΔVP30 from HeLa VP30 cells treated with the increasing doses of IFNγ for 24 h prior to infection at an MOI of 0.01. Virus titer was measured on day 2 post infection. In a separate set of experiments, cell viability of IFNγ-treated cells was measured on day 3 post-treatment by using a cell proliferation assay. Data are presented as means ± SD, and are representative of three independent experiments. **c** Relative *BTN3A3* mRNA expression in HeLa VP30 cells under *BTN3A3* knockdown. Cells were transfected with BTN3A3 siRNAs or control siRNA and then at 48 h post transfection, were treated with 10 U ml$^{-1}$ of IFNγ for 24 h. RNA was quantified by use of qRT-PCR. Data are representative of two independent experiments performed in triplicate and are presented as means ± SD of technical triplicates. **d** Cells were transfected with siRNA and then treated with IFNγ as described in **c**. At 24 h post-IFNγ treatment, cells were infected with EBOVΔVP30 at an MOI of 0.01. Virus titer was measured on day 2 post infection. Data are presented as means ± SD, and are representative of three independent experiments. (*) indicates a statistically significant difference (*p*-values of two-tailed Student's *t*-tests; *$p < 0.05$, **$p < 0.01$) from the control. **e** BTN3A3 expression in HeLa VP30 cells that stably express BTN3A3 and Hela VP30 cells treated with or without IFNγ (10 U ml$^{-1}$ for 24 h). Cells were lysed and the indicated protein expression levels were analyzed by immunoblotting. Data are representative of two independent experiments. **f** Titers of EBOVΔVP30 from HeLa VP30/BTN3A3 cells infected with EBOVΔVP30 at an MOI of 0.001. Virus titer was measured on day 3 post infection. Data are presented as means ± SD, and are representative of three independent experiments. (*) indicates a statistically significant difference (*p*-values two-tailed Student's *t*-tests; ***$p = 0.00051$) from the control. Source data are provided as a Source Data file.

whether the ISGs tested here interfere with EBOV virion formation and budding, HEK-293T cells were transfected with expression vectors for EBOV VP40, GP, and NP together with expression vectors for each ISG or GFP (negative control). BST2 (tetherin) inhibits the formation of VLPs produced by VP40 alone[18], providing a positive control for the inhibition of VP40-driven budding in our assay (note, GP was not overexpressed in the positive control because GP is known to counteract the antiviral activity of BST2[18]). As expected, BST2 substantially reduced the amount of VP40-driven VLPs (Fig. 3c, d). Of the ISGs tested, BTN3A3 appreciably decreased the level of VLPs in the cell supernatants (Fig. 3c, d). While not reducing the amount of VP40-driven VLPs in the cell supernatant, CCDC92 appreciably decreased the amount of NP incorporation into the VLPs (Fig. 3c, d). These data suggest that BTN3A3 and CCDC92 interfere with EBOV virion formation and/or budding. In each case, the overexpression of each ISG was verified by western blot analysis (Supplementary Fig. 10).

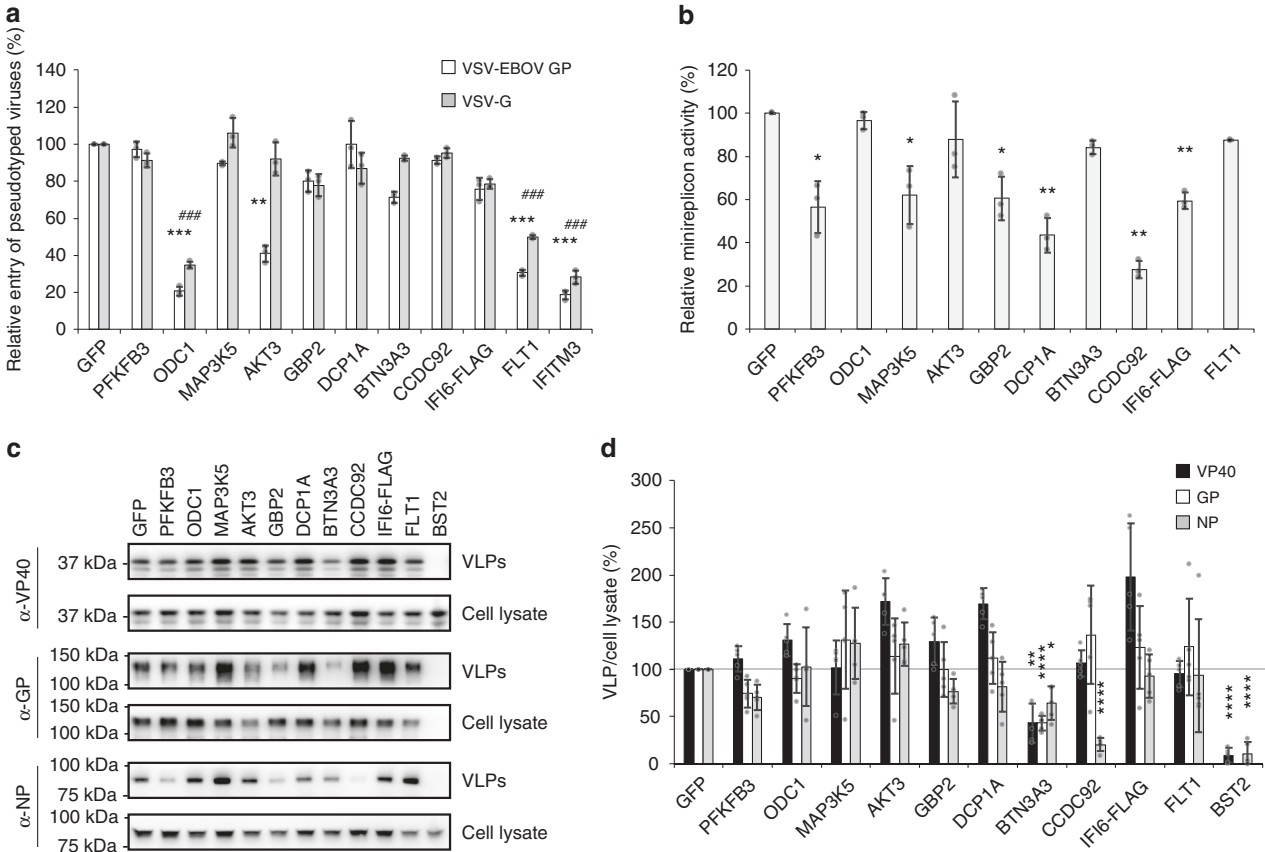

**Fig. 3 Effect of selected ISGs on EBOV cell entry, transcription/replication, and virion formation/budding. a** Relative luciferase activity in HEK-293T cells after transfection for 24 h with each of the selected ISGs and infection with VSV-EBOV GP virus (black bars) or VSV-G (gray bars) at an MOI of 0.5. IFITM3 served as a positive control for inhibition of virus entry. Data are presented as means ± SD, and are representative of three independent experiments. (*) indicates a statistically significant difference (p-values of two-tailed Student's t-tests; **p < 0.01, ***p < 0.001) for VSV-EBOV GP compared with the negative control; (#) indicates a statistically significant difference (p-values of two-tailed Student's t-tests; ###p < 0.001) for VSV-G virus compared with the negative control. **b** Relative luciferase activity in HEK-293T VP30/L cells after transfection for 48 h with vectors encoding the remaining minireplicon components, each of the 10 selected ISGs, and an internal Renilla luciferase control vector. Data are presented as means ± SD, and are representative of three independent experiments. (*) indicates a statistically significant difference (p-values of two-tailed Student's t-tests; *p < 0.05, **p < 0.01) from the control. **c, d** Representative western blot showing expression of EBOV VP40, GP, and NP in VLPs and cell lysate from transfected HEK-293T cells (**c**). The ISG BST2/tetherin, a known inhibitor of VP40-driven VLP formation, was overexpressed with VP40 alone as a positive control. Data are representative of five independent experiments. Quantification analysis of the viral proteins (VP40, GP, and NP) in VLPs following ISG overexpression compared with the GFP control was performed using ImageJ software (**d**). The ratio of the band intensity in the VLP to that in the cell lysate is indicated as a percentage. Data are presented as the mean ± SD (n = 5). (*) indicates a statistically significant difference (p-values of two-tailed Student's t-tests; *p < 0.05, **p < 0.01, ****p < 0.0001) from each of the GFP controls. Source data are provided as a Source Data file.

**CCDC92 mRNA expression after IFN treatment.** Given that the ISG, CCDC92 (Coiled-Coil Domain Containing 92), strongly inhibited EBOV infection (~600-fold reduction in virus titer; Fig. 1c), significantly inhibited virus replication/transcription (Fig. 3b), and significantly inhibited NP incorporation into virions (Fig. 3c, d), we selected CCDC92 for further mechanistic studies.

CCDC92 expression is predicted to be ubiquitous in human tissues, but especially high in the brain and male reproductive system (GeneCards database, Human Protein Atlas database). We assessed CCDC92 expression after induction by IFN treatments in various human cell lines and primary cells. Quantitative reverse transcription PCR analysis performed on two different qPCR systems showed that relative *CCDC92* mRNA expression was increased by up to threefold after IFN treatment of astrocytes and human neural stem cells, and that higher mRNA expression levels under basal cell conditions (>2-fold increase compared with IFN-untreated Huh7.0 cells) were observed in four cell types:

U-138MG (brain), HPAEC (lung), InMyoFib (intestine), and PC-3 (prostate) cells (Supplementary Fig. 11a, b).

**NP–CCDC92 interaction.** We hypothesized that CCDC92 interacts with NP and this interaction has a role in the anti-EBOV activity of CCDC92 by preventing the proper function of NP in the viral life cycle. To test this hypothesis, we first performed an immunoprecipitation assay using HEK-293T cell lysates expressing N-terminally FLAG-tagged CCDC92 (FLAG-CCDC92) with or without NP. We found that NP was efficiently co-precipitated with FLAG-CCDC92 captured by the anti-FLAG antibody (Fig. 4a). Next, we examined the cellular localization of NP and CCDC92. When only NP was expressed in HEK-293T cells, it formed small aggregates in the cytoplasm, while when only CCDC92 was expressed, it was distributed throughout the cytoplasm (Supplementary Fig. 12a). However, when NP and CCDC92 were co-expressed, the two proteins co-localized and

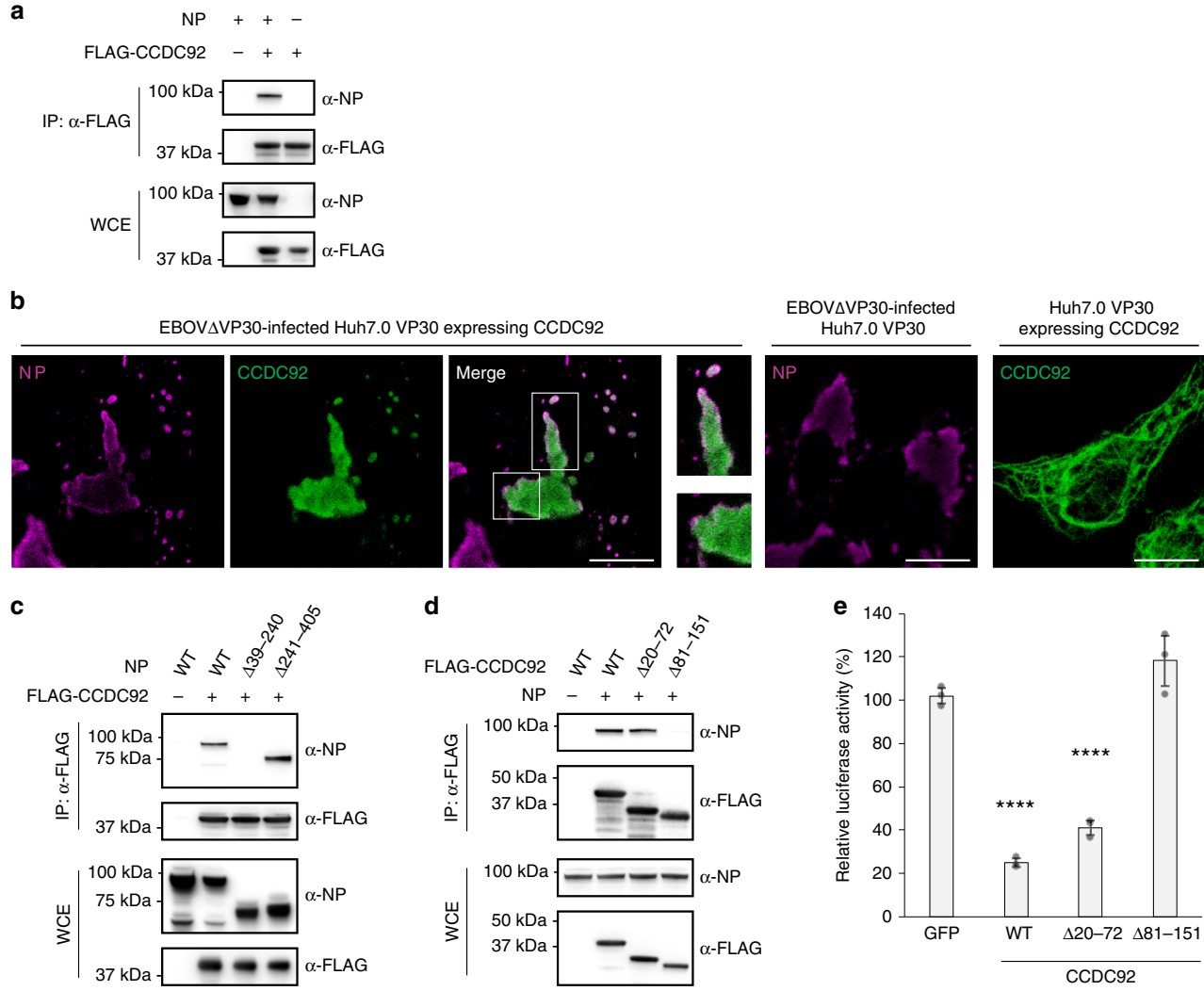

**Fig. 4 Interaction of CCDC92 with NP. a** Interaction between NP and CCDC92 in HEK-293T cells transfected with the indicated combination of expression vectors. Cell lysates were immunoprecipitated with anti-FLAG antibody followed by immunoblotting. Data are representative of three independent experiments. IP, immunoprecipitation. WCE, whole-cell extract. **b** EBOV NP (magenta) and/or CCDC92 (green) in Huh7.0 VP30 cells infected with EBOVΔVP30 24 h prior to transfection with CCDC92 expression vector were visualized with specific antibodies and analyzed by confocal microscopy. The enlarged images corresponding to the boxed areas are shown next to the original image. Data are representative of two independent experiments. Scale bars, 20 μm. **c**, **d** Mapping of the NP–CCDC92 binding regions. HEK-293T cells were transfected with expression vectors for **c** FLAG-CCDC92 and full-length NP (WT) or its deletion mutants lacking the indicated amino acid residues or **d** NP WT and FLAG-CCDC92 WT or its deletion mutants lacking the indicated amino acid residues. Cell lysates were immunoprecipitated with anti-FLAG antibody and then immunoblotted with the indicated antibodies. Data are representative of three independent experiments. IP, immunoprecipitation. **e** HEK-293T VP30 cells transfected for 24 h with expression vectors for the indicated CCDC92 constructs were infected with EBOVΔVP30-luc. Virus-driven Renilla luciferase activity was measured on day 3 post infection. Data are presented as means ± SD, and are representative of three independent experiments. (*) indicates a statistically significant difference (p-values of two-tailed Student's t-tests; ****p < 0.0001) from the control. Source data are provided as a Source Data file.

formed large aggregates in the cytoplasm in which the peak signal intensity of NP [magenta line] corresponds to that of CCDC92 [green line] (Supplementary Fig. 12a). Consistent with this observation in transfected cells expressing NP and CCDC92 only, their co-localization was also detected in EBOVΔVP30-infected Huh7.0 VP30 cells that were transfected with a CCDC92 expression vector 24 h after infection (Fig. 4b) as cells expressing CCDC92 prior to infection substantially suppressed viral protein expression given the antiviral nature of CCDC92 (Supplementary Fig. 12b, c).

Next, to identify the region of NP important for its binding to CCDC92, we generated two NP deletion mutants that lacked either the N-terminal lobe (amino acids [aa] 39–240) or the C-terminal lobe (aa 241–405) that are essential for NP

oligomerization and RNA encapsidation[37–40]. By using immunoprecipitation assays, we found that an NP mutant that lacked aa 39–240 (NP Δ39–240) did not co-precipitated with FLAG-CCDC92, whereas an NP mutant that lacked aa 241–405 (NP Δ241–405) was efficiently co-precipitated with FLAG-CCDC92 (Fig. 4c), suggesting that the N-terminal lobe of NP is required for the binding to CCDC92.

Likewise, to identify the region of CCDC92 important for binding to NP, we examined the interaction between NP and two CCDC92 deletion mutants, each lacking one of the two coiled-coil domains predicted by Jpred4. We found that a CCDC92 deletion mutant that lacked a coiled-coil domain from aa 81–151 (FLAG-CCDC92 Δ81–151) could not bind to NP, unlike the other CCDC92 deletion mutant that lacked a coiled-coil domain

from aa 20–72 (FLAG-CCDC92 Δ20–72), which could bind to NP (Fig. 4d), suggesting that the C-terminal coiled-coil domain of CCDC92 is required for the binding to NP. Consistent with these findings, CCDC92 Δ20–72 clearly co-localized with NP in EBOVΔVP30-infected Huh7.0 VP30 cells, whereas the other deletion mutant, CCDC92 Δ81–151, remained diffusely distributed throughout the cytoplasm with less clear co-localization with NP (Supplementary Fig. 12d–e).

After identifying a CCDC92 deletion mutant that no longer interacted with NP, we next determined whether this mutant still had anti-EBOV activity. In HEK-293T VP30 cells expressing wild-type CCDC92 or the deletion mutant that still interacted with NP (CCDC92 Δ20–72), we still observed inhibition of virus infection as indicated by a reduction in virus-driven luciferase expression (Fig. 4e). However, in cells expressing the deletion mutant that no longer interacted with NP (CCDC92 Δ81–151), there was a loss of CCDC92-induced anti-EBOV activity (Fig. 4e). Western blot analysis confirmed the expression of each CCDC92 construct and VP30 before infection (Supplementary Fig. 13).

**Inhibition of NP incorporation in VLPs by CCDC92**. As we previously demonstrated, CCDC92 inhibits the incorporation of NP into VP40-driven VLPs (Fig. 3c, d). Incorporation of NP into VLPs requires the interaction of NP with VP40, which transports NP to the cell surface[41,42]. When NP and VP40 were co-expressed by protein expression vectors in HEK-293T cells, their co-localization was observed on the cell surface (Fig. 5a); however, in the presence of CCDC92, NP co-localized with CCDC92 and NP–VP40 co-localization was not observed (Fig. 5a).

This result led us to examine whether CCDC92 inhibits the interaction between NP and VP40. The binding of NP to VP40 was detected by using an immunoprecipitation assay with HEK-293T cells expressing N-terminally FLAG-tagged NP (FLAG-NP) and VP40 (Fig. 5b). When CCDC92 was co-expressed with FLAG-NP and VP40, the binding of NP to VP40 was significantly disrupted by nearly 90% (Fig. 5b and Supplementary Fig. 14a). Surprisingly, CCDC92 Δ20–72, the deletion mutant that still interacted with NP, did not inhibit the interaction between NP and VP40, and neither did the other deletion mutant, CCDC92 Δ81–151 (Fig. 5b and Supplementary Fig. 14a). Moreover, neither CCDC92 deletion mutant inhibited the incorporation of NP into VP40-driven VLPs (Supplementary Fig. 14b). Interestingly, the CCDC92 deletion mutant (CCDC92 Δ20–72) that still interacted with NP in immunoprecipitation assays (Fig. 5b) and still had anti-EBOV activity in infected cells (Fig. 4e) was incorporated into the VLPs (Supplementary Fig. 14b).

**Inhibition of EBOV transcription by CCDC92**. To further examine the inhibition of EBOV transcription by CCDC92, we assessed the mRNA levels of viral *NP* in EBOVΔVP30-infected cells. NP is the most abundant mRNA transcript, and the most abundant viral protein in EBOV-infected cells[43]. Compared with the empty vector control cells, *NP* mRNA levels were significantly reduced by nearly 90% in infected cells that overexpressed CCDC92 (Fig. 5c). As expected, BTN3A3 did not affect *NP* mRNA levels (Fig. 5c). In the same experimental setting, we confirmed the downstream reduction of viral protein levels by CCDC92 in infected cells compared with either the empty vector control or BTN3A3 (Fig. 5d).

To determine further whether CCDC92 inhibits viral transcription, we also assessed *NP* mRNA levels in EBOVΔVP30-infected cells in the presence or absence of cycloheximide (CHX), an inhibitor of protein synthesis. In cells treated with CHX (10–150 μg ml$^{-1}$), the expression of VP30 in HEK-293T cells was reduced, but still detectable by western blot analysis after 24 h

of treatment (Supplementary Fig. 15a). In HEK-293T VP30 cells infected with EBOVΔVP30, CCDC92 still significantly reduced *NP* mRNA levels at 8 h and 24 h post infection even in cells treated with CHX (50 μg ml$^{-1}$) (Supplementary Fig. 15b; top panel without CHX treatment and bottom panel with CHX treatment). From these data, we can conclude that CCDC92 inhibits viral transcription. In a separate set of experiments, we confirmed the expression both CCDC92 and BTN3A3 in transfected cells prior to CHX treatment (Supplementary Fig. 15c).

Furthermore in the minireplicon assay, both wild-type CCDC92 and the deletion mutant that still interacted with NP (CCDC92 Δ20–72) significantly inhibited luciferase activity in the minireplicon assay, whereas the other CCDC92 deletion mutant that cannot interact with NP (CCDC92 Δ81–151) had no inhibitory effect (Fig. 5e); an antiviral effect similar to the one observed in EBOVΔVP30-infected cells (Fig. 4e). Western blot analysis confirmed the expression of each CCDC92 construct as well as that of the viral proteins in the minireplicon assay (Supplementary Fig. 16).

**Mechanism of CCDC92 inhibition in viral transcription**. The RNA genome is tightly encapsidated by NP oligomers and this encapsidated RNA serves as a template for EBOV replication/transcription[44,45]. NP oligomerization and RNA binding occur simultaneously and synergistically[46], and these events are mediated by the N-terminal region of NP that includes the N-terminal arm (aa 19–38) and NP lobes (aa 39–405) including the C-terminal lobe of NP necessary for its interaction with CCDC92 (Fig. 4c). To further understand the mechanism by which CCDC92 inhibits NP function, we assessed the impact of CCDC92 on NP oligomerization. When we co-expressed C-terminally FLAG- and His-tagged NP (NP-FLAG and NP-His) with and without CCDC92 in HEK-293T cells and immunoprecipitated the cell lysates with anti-FLAG antibody, we found no difference in the interaction between the two tagged versions of NP, suggesting that CCDC92 does not interfere with NP oligomerization (Supplementary Fig. 17).

Next, to examine whether CCDC92 inhibits NP–RNA binding, we performed an RNA-immunoprecipitation assay (RNA-IP) to quantify the amount of NP bound to viral RNA. We found no significant difference between the amount of RNA bound to NP in the presence or absence of CCDC92, suggesting CCDC92 does not interfere with the interaction between NP and viral RNA (Supplementary Fig. 18a, b).

These observations led us to examine whether CCDC92 interacts with a complex of NP and viral RNA. HEK-293T cells expressing FLAG-CCDC92 and EBOV minireplicon RNA with or without NP were analyzed by using RNA-IP, as described above. In this case, we found that EBOV RNA was precipitated with FLAG-CCDC92 along with NP, suggesting the formation of a CCDC92–NP–RNA complex (Fig. 6a, b).

Viral replication/transcription in EBOV-infected cells takes place in viral inclusion bodies, which consist of viral RNA, NP, VP35, VP30, and L[47–50]. The finding that CCDC92 binds to an NP–RNA complex and inhibits viral RNA synthesis led us to hypothesize that the inhibition of NP function by CCDC92 may be associated with viral inclusion bodies. Therefore, we examined whether CCDC92 co-localizes with proteins that form inclusion bodies: NP, VP35, VP30, and HA-tagged L (L-HA). HEK-293T cells were transfected with vectors expressing the viral proteins (NP, VP35, VP30, and L-HA) and co-localization was detected by use of immunofluorescence microscopy using specific antibodies against each viral protein or an antibody against the HA-tag in the case of the viral L protein. When co-expressed, CCDC92 co-localized with all four viral proteins involved in inclusion body formation (Fig. 6c). The

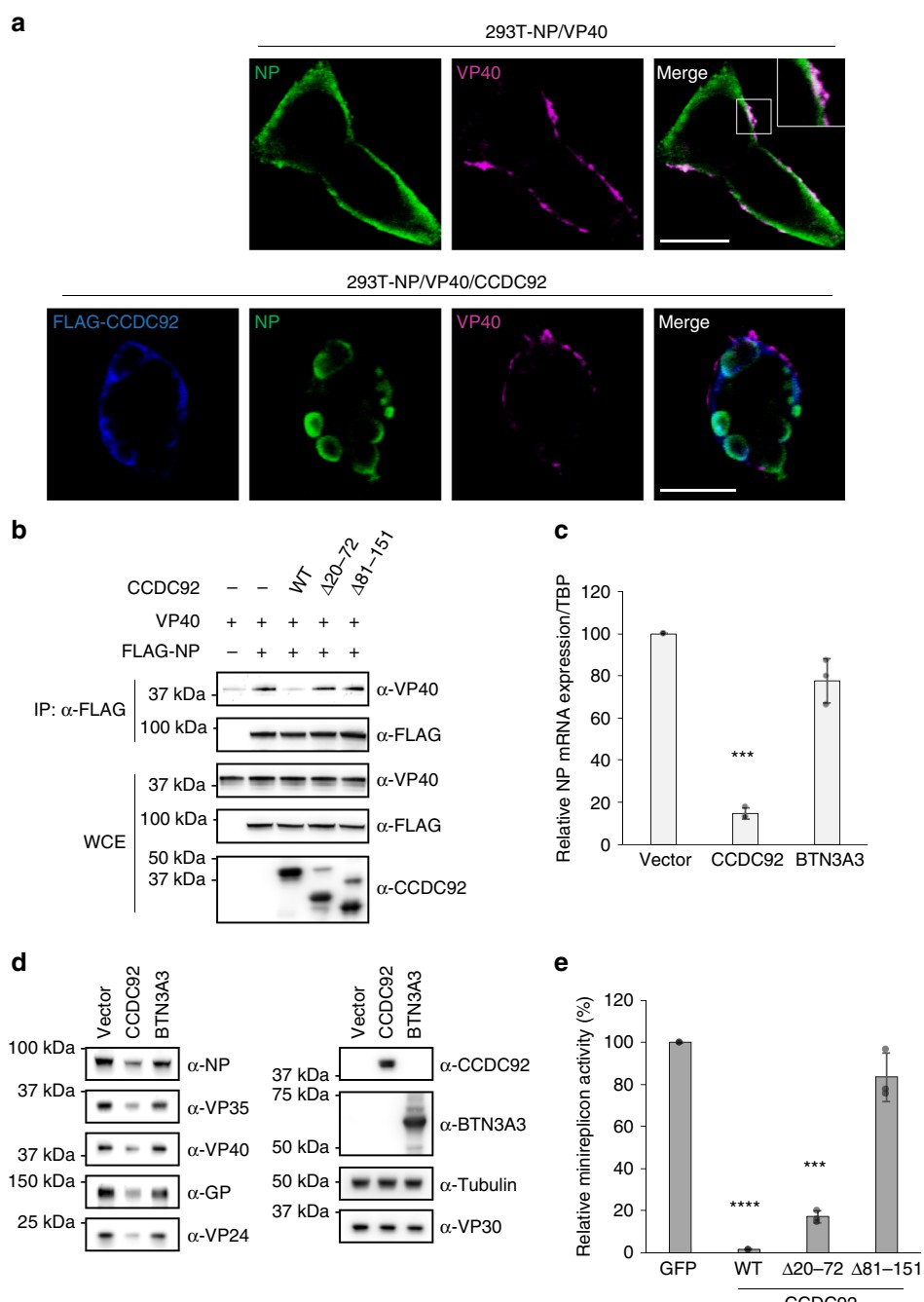

**Fig. 5 Inhibition of the NP–VP40 interaction and viral transcription by CCDC92. a** EBOV NP (green) and VP40 (magenta) with or without CCDC92 (blue) in HEK-293T cells transfected with expression vectors for NP and VP40 together with or without FLAG-CCDC92 were visualized with specific antibodies and analyzed by confocal microscopy. The inset is the enlarged image corresponding to the boxed area. Data are representative of two independent experiments. Scale bars, 20 μm. **b** Interaction between NP and VP40 in the presence of CCDC92 or its deletion mutants in HEK-293T cells transfected with the indicated combination of expression vectors. Cell lysates were immunoprecipitated with anti-FLAG antibody followed by immunoblotting. Data are representative of three independent experiments. IP, immunoprecipitation. WCE, whole-cell extract. **c** Relative *EBOV NP* mRNA expression in HEK-293T cells after overexpression of CCDC92 or BTN3A3. Cells were transfected with expression vectors for CCDC92 or BTN3A3, or with an empty control vector for 24 h prior to infection with EBOVΔVP30 at an MOI of 3.0. At 24 h post infection, RNA was extracted and then quantified by qRT-PCR. The results are normalized to *TBP* mRNA expression, and the data are presented as percentages ± SD ($n = 3$). (*) indicates a statistically significant difference (*p*-values of two-tailed Student's *t*-tests; ***$p < 0.001$) from the control. **d** Cells were transfected and infected as described in **c**. At 24 h post infection, cells were lysed and the indicated protein expression levels were analyzed by immunoblotting. **e** Relative luciferase activity in HEK-293T VP30 cells after transfection for 48 h with vectors required for the minireplicon assay, each of the indicated genes, and an internal Renilla luciferase control vector. Data are presented as means ± SD, and are representative of three independent experiments. (*) indicates a statistically significant difference (*p*-values of two-tailed Student's *t*-tests; ***$p < 0.001$, ****$p < 0.0001$) from the control. Source data are provided as a Source Data file.

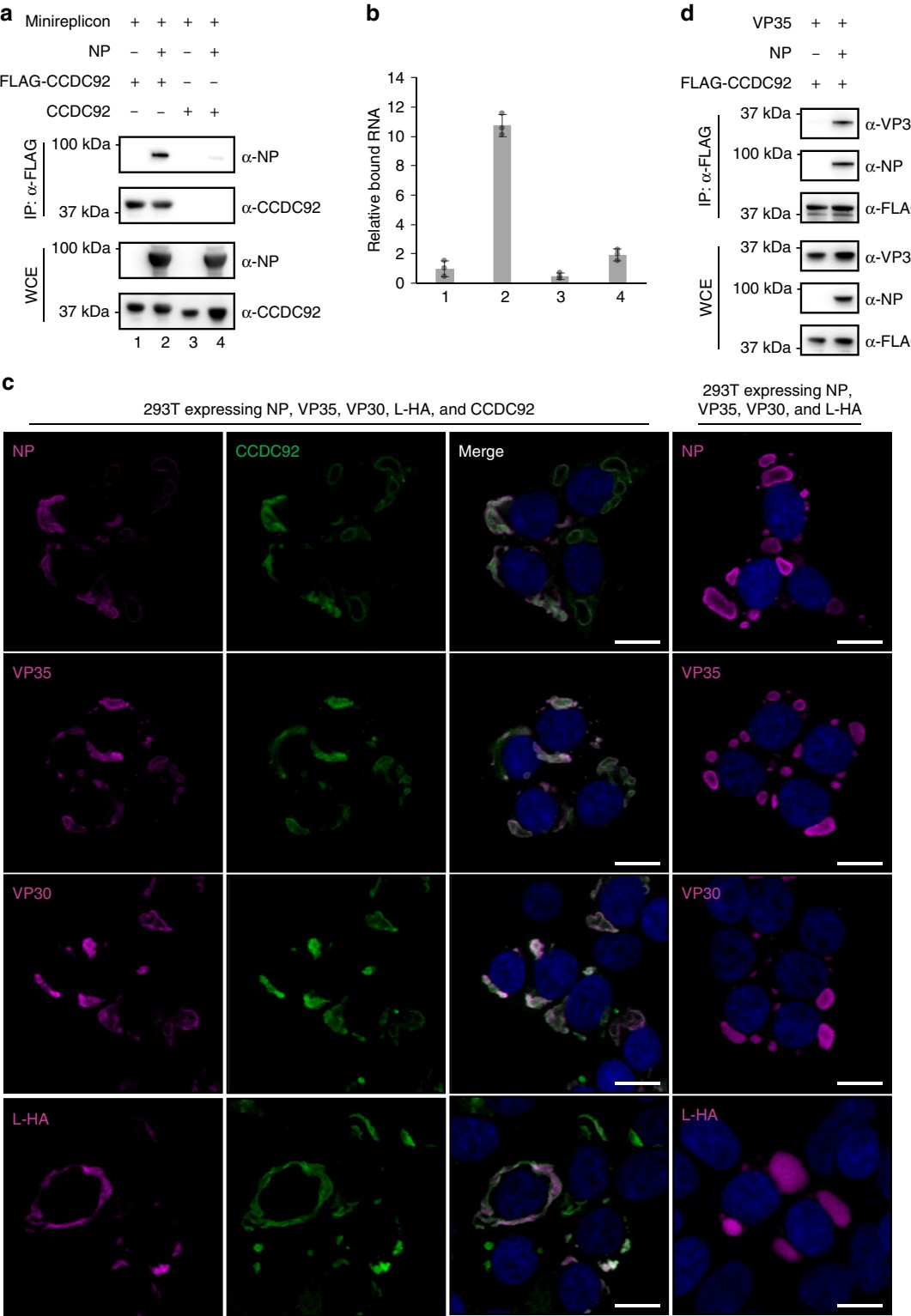

co-localization of NP and CCDC92 with viral proteins that are involved in inclusion body formation was also observed with the CCDC92 deletion mutant that interacts with NP (CCDC92 Δ20–72; Supplementary Fig. 19a). But the deletion mutant that does not interact with NP and does not have antiviral activity (CCDC92 Δ81–151) showed more diffuse cytoplasmic staining and less clear co-localization with NP in these inclusion body-like structures (Supplementary Fig. 19b), similar to the pattern observed in infected cells (Supplementary Fig. 12d–e).

We also showed the formation of a complex comprising CCDC92, NP, and VP35 (a viral polymerase co-factor) by co-immunoprecipitation that was dependent on NP for the incorporation of VP35 into this complex (Fig. 6d). Taken together, these results suggest that the interaction of CCDC92 with NP bound to RNA and VP35, which may occur within inclusion bodies or before NP begins inclusion body formation, most likely disrupts functional viral replication/transcription protein complexes.

**Fig. 6 Interaction of CCDC92 with NP and viral-like RNA. a, b** Relative amounts of EBOV minireplicon RNA precipitated with FLAG-CCDC92 along with or without NP. HEK-293T cells were transfected with the indicated combination of vectors shown in **a**. Cell lysates were immunoprecipitated with anti-FLAG antibody followed by immunoblotting (**a**). Quantification of EBOV minireplicon RNA was performed by use of qRT-PCR (**b**). The amount of EBOV minireplicon RNA precipitated was normalized against the amount of EBOV minireplicon RNA in the whole-cell lysates for each corresponding sample. The numbers under the blots (**a**) correspond to the numbers under the bar graph (**b**). Data are representative of two independent experiments performed in triplicate. Error bars in **b** indicate SD of technical triplicates. IP, immunoprecipitation. WCE, whole-cell extract. **c** EBOV proteins (magenta) with or without CCDC92 (green) in HEK-293T cells transfected with expression vectors for EBOV proteins (NP, VP35, VP30, and L-HA) together with or without CCDC92 were visualized with specific antibodies and analyzed by confocal microscopy. Nuclei were visualized with DAPI (blue). Data are representative of two independent experiments. Scale bars, 20 μm. L-HA, HA-tagged L. **d** Interaction between VP35 and CCDC92 in the presence or absence of NP in HEK-293T cells transfected with expression vectors for VP35 and FLAG-CCDC92 together with or without NP. Cell lysates were immunoprecipitated with anti-FLAG antibody followed by immunoblotting. Data are representative of three independent experiments. IP, immunoprecipitation. Source data are provided as a Source Data file.

## Discussion

Our ISG library screen identified cellular proteins involved in the host restriction of EBOVΔVP30 infection; the potential of these proteins to inhibit wild-type EBOV still needs to be investigated. Overexpression of 21 ISGs caused a statistically significant reduction in luciferase expression levels ($p$-value ≤ 0.05) and overexpression of one additional ISG (IFI6; $p$-value = 0.075) caused a marked reduction in luciferase levels (Fig. 1a, b). Several of these 22 ISGs have established roles in the innate immune response to viruses, including interferon regulatory factor 1 (IRF1)[15,51,52], interferon regulatory factor 7 (IRF7)[53], interferon induced with helicase c domain 1 (IFIH1/MDA5)[54], and guanylate binding protein 2 (GBP2)[55]. In addition, overexpression of interferon alpha-inducible protein 6 (IFI6) and CD9 has been shown to inhibit yellow fever virus, West Nile virus, and dengue virus[15,19,20,28,56]. Moreover, interferon-induced transmembrane protein 3 (IFITM3) has been shown to inhibit EBOV entry[33], demonstrating the potency of our screen to reveal ISGs with antiviral properties.

We chose the top 10 ISGs (summarized in Supplementary Table 1) that had the greatest impact on EBOVΔVP30 infection without causing non-specific effects on cellular function or viability for in-depth analyses. These ISGs were tested for their ability to inhibit EBOVΔVP30 growth kinetics. All 10 ISGs significantly attenuated EBOVΔVP30 growth on day 6 post infection (Fig. 1c) and could function in even primary cells (Supplementary Fig. 3). Endogenous BTN3A3 induced by IFNγ was shown to contribute to this anti-EBOV activity (Fig. 2 and Supplementary Fig. 5). These findings suggest that the top 10 ISGs identified in our screen may represent targets for anti-EBOV therapy.

The ISG CCDC92 was of particular interest because it inhibited viral transcription/replication by more than 70% in a minireplicon assay (Figs. 3b and 5e) and reduced the amount of NP incorporated into VLPs by nearly 80% with no significant effect on the amount of VP40 or GP incorporated into the VLPs (Fig. 3c, d and Supplementary Fig. 14b). Our data suggest that the inhibitory effects of CCDC92 on these two steps in the virus life cycle stem from its interaction with NP (Fig. 4a). Moreover, by binding to and retaining NP in the cytoplasm, CCDC92 inhibits the interaction between NP and VP40 (Fig. 5a, b), which is essential for the transport of NP to the cell surface[42].

In regards to viral replication/transcription, we were able to confirm that the overexpression of CCDC92 inhibited viral mRNA transcription and the overall levels of viral proteins in infected cells (Fig. 5c, d and Supplementary Fig. 15). We found that the N-terminal lobe of NP (aa 39–240), which composes an RNA-binding groove essential for NP oligomerization and RNA encapsidation[37–40], is necessary for the interaction with CCDC92 (Fig. 4c); however CCDC92 does not inhibit the interaction of NP with itself or the interaction of NP with viral RNA (Supplementary Figs. 17 and 18, respectively). Instead, our data suggest that CCDC92 forms a complex with viral RNA-bound NP (Fig. 6). It remains unclear whether CCDC92 can target the incoming genomic RNA bound to NP and other viral proteins of the viral transcription/replication machinery (i.e., VP35, VP30, VP24, and L[8,10,57,58]) after uncoating in the cytoplasm, where RNA/NP/CCDC92 complex formation occurs. However, CCDC92 clearly inhibits viral transcription, which is mediated through the C-terminal coiled-coil domain of CCDC92 (aa 81–151) that is necessary for the interaction with NP (Figs. 4d, 5e, and Supplementary Fig. 12d, e), as supported by the finding that CCDC92 viral mRNA and protein expression in infected cells (Fig. 5c, d and Supplementary Fig. 15).

EBOV replication and transcription occur in inclusion bodies, which have been observed at about 10 h post infection[47,59]. We showed the co-localization of CCDC92 within these inclusion bodies formed by NP, VP35, VP30, and the viral RNA polymerase, L through NP (Fig. 6c and Supplementary Fig. 19a, b). Furthermore, we found an indirect interaction between CCDC92 and VP35 through NP (Fig. 6d). VP35 is a polymerase co-factor that forms an active polymerase complex with L, bridging NP and L to allow the polymerase complex to access the RNA genome[8,57]. Taken together, our results imply that CCDC92 inhibits EBOV transcription by binding to the polymerase complex via NP bound to viral RNA in the inclusion bodies.

*CCDC92* mRNA expression was induced by IFN treatment in astrocytes and human neural stem cells, and relatively high mRNA expression levels under basal cell conditions were observed in four cell types from brain, lung, intestine, and prostate (Supplementary Fig. 11). Although EBOV antigens have been described in these tissues, non-significant or mild inflammation during EBOV infection in these tissues has been reported[29,60]. This may suggest that CCDC92 has a role in cell susceptibility to EBOV or tissue tropism during infection.

In summary, we identified several ISGs with previously unknown antiviral activity against EBOV whose overexpression significantly inhibited EBOVΔVP30 growth. These findings may help identify targets for the development of antivirals to combat EBOV infection.

## Methods

**Cells**. Vero VP30 cells (African green monkey kidney cells stably expressing EBOV VP30) were established as previously described[26]. Vero VP30 cells were grown in Eagle's minimal essential medium (MEM) supplemented with 10% fetal bovine serum (FBS), L-glutamine, vitamins, nonessential amino acid solution, and antibiotics. HEK-293T VP30 cells (human embryonic kidney cells stably expressing the SV40 T-antigen and EBOV VP30), HEK-293T VP30/L cells (human embryonic kidney cells stably expressing EBOV VP30 and EBOV L), and Huh7.0 VP30 (human hepatocarcinoma cells stably expressing EBOV VP30) were established in a similar manner as Vero VP30 cells[26].

HeLa VP30 cells (human cervical cancer cells stably expressing EBOV VP30) and HeLa VP30/BTN3A3 (human cervical cancer cells stably expressing EBOV VP30 and BTN3A3) were generated as follows: a cDNA fragment encoding EBOV

VP30 or BTN3A3 was cloned into the murine leukemia virus (MLV)-based retroviral vectors pMXs-IRES-Neo (pMXs-IN) and pMXs-IRES-Puro (pMXs-IP) (Cell Biolabs), respectively. To generate the retrovirus, Plat-GP cells (Cell Biolabs) were co-transfected with pMXs-IN encoding EBOV VP30 or pMXs-IP encoding BTN3A3 along with an expression vector for VSV-G by using Lipofectamine 2000 (Invitrogen). Two days later, the culture supernatants containing the retroviruses were collected, clarified through 0.45-μm-pore filters, and then used to infect Hela cells. Stable cells were selected with 500 μg ml⁻¹ G418 (InvivoGen) with or without 2 μg ml⁻¹ puromycin (InvivoGen).

All HEK-293T cell lines (wild-type, VP30, or VP30/L), Huh7.0 cell lines (wild-type or VP30), and HeLa cell lines (wild-type, VP30, or VP30/BTN3A3) were grown in high-glucose Dulbecco's modified Eagle's medium (DMEM) containing 10% FBS, L-glutamine, antibiotics, and appropriate selection drugs. All cells were maintained at 37 °C and 5% $CO_2$.

**Viruses**. EBOVΔVP30 expressing GFP (EBOVΔVP30-GFP) or Renilla luciferase (EBOVΔVP30-luc) instead of the viral *VP30* gene were propagated in Vero VP30 cells using propagation medium similar to cell growth medium, but supplemented with only 2% FBS[26]. The use of EBOVΔVP30 viruses in BSL-2 containment at the University of Wisconsin is approved by the Institutional Biosafety Committee, the NIH, and the CDC. A VSV possessing the firefly luciferase gene in place the VSV-*G* gene and pseudotyped with EBOV GP (VSV-EBOV GP) and a control luciferase-expressing VSV containing only VSV-G (VSV-G) were prepared[7]. To generate VSV-EBOV GP, HEK-293T cells were transfected for 24 h with an EBOV GP expression vector and then were infected with VSV-G at an MOI of >1.0. Supernatant was collected at 24 h post infection, clarified through 0.45-μm-pore filters, and then used for the entry assay.

**ISG screen**. For the primary screen, we obtained a cDNA library of 389 ISGs that was previously described[15]. HEK-293T VP30 cells were seeded into poly D-lysine-coated 24-well plates (Bio-Coat). Cells were transfected in duplicate for 24 h with 0.25 μg of each ISG protein expression vector or a control vector encoding a red fluorescent protein by using Lipofectamine 2000 (Life Technologies) and were then infected with EbolaΔVP30-luc virus at an MOI of 1.0. Two days post infection, cells were lysed with 300 μl of 1× Glo-lysis buffer (Promega) and analyzed for Renilla luciferase activity by using the Renilla luciferase assay system (Promega) according to the manufacturer's instructions. Expression of the top 10 ISGs selected for further characterization was confirmed by western blotting (Supplementary Fig. 20). The IFI6 gene was C-terminally tagged with FLAG, inserted into the LentiX and pCAGGS vectors, and then used for the assays.

**Assessment of viral titers**. To validate the antiviral activity of the candidate ISGs, the ISGs were cloned into the pCAGGS protein expression vector[26]. We then transfected HEK-293T VP30 cells for 24 h with the pCAGGS-ISG expression vectors (0.25 μg each) or a control vector encoding GFP by using TransIT-LT1 (Mirus). Cells were then infected with EBOVΔVP30-GFP virus at an MOI of 0.0001. Supernatants were harvested on days 3 and 6 post infection, and viral titers were determined by means of focus forming assay[26]. Briefly, 10-fold dilutions of the virus in the supernatant were absorbed to confluent Vero VP30 cells for 1 h at 37 °C. Unbound virus was removed by washing with culture medium. The cells were then overlaid with medium containing 1.5% methylcellulose (Sigma). After a 7-day incubation, the cells were fixed with 10% buffered formaldehyde, and permeabilized with 0.25% Triton X-100 (Sigma) for 15 min. The cells were then incubated overnight at 4 °C with a mouse anti-VP40 monoclonal antibody (1:1000, cl. 6), followed by a 2-h incubation with an anti-mouse secondary antibody coupled to horseradish peroxidase (1:2000, G21040, Life Technologies). After washing with PBS, the cells were incubated with 3,3-diaminobenzidine tetrahydrochloride (Sigma) in PBS. The reaction was stopped by rinsing with water.

**Assessment of cell entry**. To examine cell entry mediated by EBOV GP, HEK-293T cells were seeded into 24-well tissue culture plates and transfected for 24 h with individual ISG protein expression vectors (0.25 μg), a negative control vector (0.25 μg), or a vector encoding IFITM3 (0.25 μg) by using TransIT-LT1 (Mirus). A neutralizing monoclonal antibody against the VSV-G protein (cl. I-1) was used to abolish the background infectivity of parental VSV-G virus in the virus stock of VSV-EBOV GP. Cells were then infected with VSV-EBOV GP virus or VSV-G virus at an MOI of 0.5. Twenty-four hours later, cells were lysed with 300 μl of 1× Glo-lysis buffer (Promega) and analyzed for firefly luciferase activity by using the Steady-Glo luciferase assay system (Promega) according to the manufacturer's instructions.

**Assessment of viral polymerase activity**. To examine viral polymerase activity, a luciferase activity-based EBOV minireplicon assay was performed as described previously[61]. Briefly, HEK-293T VP30/L cells were seeded into 24-well tissue culture plates and co-transfected with two EBOV protein expression vectors [pCAGGS EBOV NP (0.05 μg) and pCAGGS EBOV VP35 (0.05 μg)], an EBOV minireplicon vector encoding firefly luciferase under the control of the RNA polymerase I promoter (EBOV pol-luc, 0.2 μg), and a Renilla luciferase protein expression vector (0.002 μg) as an internal control. To test the effect of ISG

expression on EBOV polymerase activity, cells were co-transfected with vectors encoding individual ISGs (0.3 μg). Two days after transfection, cells were lysed in 300 μl of 1× Glo-lysis buffer (Promega) and analyzed for firefly and Renilla luciferase activities by using the dual-luciferase reporter assay system (Promega) according to the manufacturer's instructions.

**Assessment of virus-like particle (VLP) formation**. To evaluate the effect of ISG expression on VP40-mediated budding or the formation of VLPs, 293T cells were seeded in 12-well plates for 24 h and then co-transfected with three Ebola viral protein expression vectors [pCAGGS EBOV VP40 (0.1 μg), pCAGGS EBOV GP (0.1 μg), pCAGGS EBOV NP (0.1 μg)], and an expression vector encoding an ISG (0.3 μg) by using TransIT-LT1 (Mirus). As a positive control for inhibition of VLP formation, cells were transfected with pCAGGS EBOV VP40 (0.3 μg) and pCAGGS BST2/tetherin (0.3 μg) only. Two days after transfection, cell supernatants containing VLPs were harvested and centrifuged through 20% sucrose for 1.5 h at 27,000 rpm. Cell lysates were also harvested and, in parallel with the VLP pellets, resuspended in 1× Laemmli sample buffer containing 5% β-mercaptoethanol for subsequent western blot analysis with mouse anti-EBOV VP40 (1:1000, cl. 6) and anti-EBOV GP (1:1000, 254/3.12) antibodies, as well as a rabbit anti-EBOV NP (1:1000, R5071) antibody.

**Western blot analysis**. Cells were washed with PBS and treated with lysis buffer (1% NP-40, 50 mM Tris-HCl [pH 7.4], 150 mM NaCl, 0.25% sodium deoxycholate, with/without 0.1% SDS) containing protease inhibitor cocktail (Sigma). Cell lysates were mixed with an equal volume of 2× Laemmli sample buffer (Bio-Rad) containing 5% β-mercaptoethanol and then incubated at 95 °C for 10 min. Samples were resolved by SDS-PAGE onto a 4–20% SDS-polyacrylamide gel (Life Technologies). After electrophoresis, proteins were transferred onto a polyvinylidenedifluoride membrane (Life Technologies) and blocked for 1 h at room temperature with 5% skim milk/TBS solution containing 0.05% Tween-20 (TBS-T; Sigma). Membranes were incubated overnight at 4 °C with the primary antibodies against EBOV VP40 (1:1000, cl. 6), EBOV GP (1:1000, cl. 254/3.12), EBOV NP (1:1000, R5071), EBOV VP35 (1:1000, cl. 1), EBOV VP30 (1:1000, cl. 3), EBOV VP24 (1:1000, cl. 21-5.2.5), alpha-tubulin (1:1000, ab7291, Abcam), PFKFB3 (1:1000, 13132, Cell signaling technology), ODC1 (1:1000, TA501546, Origene), MAP3K5 (1:1000, 8662, Cell signaling technology), AKT3 (1:1000, 3788, Cell signaling technology), GBP2 (1:1000, TA500657, Origene), DCP1A (1:1000, D5444, Sigma), BTN3A3 (1:1000, HPA007904, Sigma), CCDC92 (1:1000, ab104028, Abcam), FLT1 (1:1000, TA303515, Origene), FLAG tag (1:2000, M2, Sigma), and His tag (1:1000, D3I1O, Cell signaling technology). Membranes were then incubated for 1 h with the following secondary antibodies coupled to horseradish peroxidase: a goat anti-mouse IgG (1:3000, G21040, Life Technologies), a goat anti-rabbit IgG (1:3000, G21234, Life Technologies), or a goat anti-rat IgG (1:3000, A10549, Life Technologies). Bound antibody was detected with Super-Signal Pico, Dura, or Femto chemiluminescence reagent (Thermo Scientific) and a FluorChem HD2 imager (Alpha Innotech).

**Cell viability assay**. Cell viability was assessed by using the Non-Radioactive Cell Proliferation Assay (Promega) or Cell Titer-Glo (Promega) according to the manufacturer's protocol. Absorbance and luminescence were measured on a Tecan M1000 plate reader.

**Transfection of siRNA**. Human BTN3A3 siRNAs (Hs_BTN3A3_1 and Hs_BTN3A3_4) and a negative control siRNA (All Star Negative Control siRNA) were purchased from Qiagen. Cells were seeded in 24-well tissue culture plates at $1.0 \times 10^5$ cells per well 1–2 h prior to transfection. Each siRNA (5 pmol) was mixed with 1.5 μl of Lipofectamine RNAiMAX reagent (Invitrogene) in 50 μl of Opti-MEM (Life Technologies) and then incubated for 10 min. The mixture was then added to the wells. After a 48-h incubation, the culture medium was replaced with fresh medium with or without IFNγ (Roche). Knockdown efficiency was evaluated at 72 h post transfection by use of qRT-PCR. EBOVΔVP30 infection was performed at an MOI of 0.01 at 72 h post transfection and then cells were cultured in the presence or absence of IFNγ. Supernatants were harvested on day 2 post infection, followed by virus titration.

**Quantitative reverse transcription PCR (qRT-PCR)**. RNA extraction was carried out with an RNeasy Mini Kit (Qiagen). Approximately 400–600 ng of total RNA was reverse-transcribed into cDNA using a QuantiTect Reverse Transcription Kit (Qiagen) (for EBOV *NP*, oligo dT primer was used). The cDNA was amplified and analyzed by using PowerUp SYBR Green Master Mix (Life Technologies) on QuantStudio 6 Flex (Applied Biosystems) following the manufacturer's protocol. The prime sequences were as follows: 5′- TTGACAGCAGGTCTGTCCGTTCAA-3′ (forward primer targeting EBOV *NP* gene), 5′- AACAACTGCTTCAAAGGC CTGTA-3′ (reverse primer targeting EBOV *NP* gene), 5′- ATCTCCTACCTTGG CGACCT-3′ (forward primer targeting EBOV minireplicon), and 5′- GTTTGTGA TGCCATCCGACG-3′ (reverse primer targeting EBOV minireplicon). Primers targeting the TATA box-binding protein (*TBP*) gene, which was used as an internal control gene, were designed and synthesized commercially by Qiagen. The results

are expressed as fold-changes normalized to *TBP* gene expression by using the ΔΔCt method.

**Immunofluorescence assay**. HEK-293T cells and Huh7.0 VP30 were fixed with 4% paraformaldehyde for 15 min at room temperature and permeabilized with 0.25% Triton X-100 for 15 min. After being blocked with 1% BSA in PBS, the cells were incubated with the following primary antibodies: a mouse anti-NP monoclonal antibody (1:200, cl. 7-47-18), a rabbit anti-NP polyclonal antibody (1:200, R5071), a mouse anti-GP monoclonal antibody (1:200, cl. 226/8.1), a mouse anti-VP40 monoclonal antibody (1:200, cl. 6), a mouse anti-VP30 monoclonal antibody (1:200, cl. 19), a mouse anti-VP35 monoclonal antibody (1:200, cl. 5-82-6.3), a rat anti-HA monoclonal antibody 3F10 (1:200, Roche), a mouse anti-FLAG monoclonal antibody M2 (1:200, Sigma), a rat anti-FLAG monoclonal antibody (1:200, SAB4200071, Sigma), or a rabbit anti-CCDC92 polyclonal antibody (1:200, ab104028, Abcam). Cells were washed three times with PBS and incubated with 4′,6-diamidino-2-phenylindole (DAPI) and the following secondary antibodies: a goat anti-mouse IgG–Alexa Fluor 488 (1:200, A11029, Life Technologies), a goat anti-rabbit IgG–Alexa Fluor 488 (1:200, A11034, Life Technologies), a goat anti-mouse IgG–Alexa Fluor 546 (1:200, A11030, Life Technologies), a goat anti-rabbit IgG–Alexa Fluor 546 (1:200, A11035, Life Technologies), a goat anti-rat IgG–Alexa Fluor 546 (1:200, A11081, Life Technologies), or a goat anti-rat IgG–Alexa Fluor 633 (1:200, A21094, Life Technologies). Stained samples were analyzed using an LSM 510 META confocal microscope (Carl Zeiss) with ZEN 2009 software (Carl Zeiss). HA-tagged *L* gene was constructed by inserting an HA-tag (YPYDVPDYA) between nucleotides 5115 and 5116 (amino acids P1705 and Q1706) in a putative linker region[47,49] using a PCR-based approach.

**Co-immunoprecipitation assay**. The CCDC92 gene was N-terminally tagged with FLAG and inserted into a pCAGGS expression vector. The deletion mutants of CCDC92 and NP were constructed using a PCR-based approach. HEK-293T cells were transfected with the indicated expression vectors using TransIT-LT1 (Mirus). Two days after transfection, cells were washed once with PBS and treated for 30 min with lysis buffer (0.1% SDS, 1% NP-40, 50 mM Tris-HCl [pH 7.4], 150 mM NaCl, 0.25% sodium deoxycholate) containing protease inhibitor cocktail (Sigma) to prepare whole-cell extract (WCE). Cell extracts were mixed with pre-conjugated anti-FLAG M2 magnetic beads (Sigma) and incubated on a rotator at 4 °C overnight. The next day, the magnetic beads were washed three times with lysis buffer. The bound proteins were eluted with excess FLAG peptide (Sigma) and analyzed by SDS-PAGE followed by immunoblotting with the indicated antibodies.

**RNA-immunoprecipitation assay**. HEK-293T cells were transfected with the EBOV minireplicon vector and the indicated protein expression vectors by using TransIT-LT1 (Mirus). Two days after transfection, cells were washed once with PBS and treated for 30 min with lysis buffer (0.1% SDS, 1% NP-40, 50 mM Tris-HCl [pH 7.4], 150 mM NaCl, 0.25% sodium deoxycholate) containing protease inhibitor cocktail (Sigma). Cell extracts were mixed with pre-conjugated anti-FLAG M2 magnetic beads (Sigma) and incubated on a rotator at 4 °C overnight. The next day, the magnetic beads were washed three times with lysis buffer. The bound proteins were eluted with excess FLAG peptide (Sigma). RNA was extracted from cell lysates with the RNeasy Mini Kit (Qiagen), reverse-transcribed using QuantiTect Reverse Transcription Kit (Qiagen), and then analyzed by qRT-PCR.

**Statistics and reproducibility**. Statistical analysis was conducted using R v3.1.1 and Microsoft Excel 2016. To determine which ISGs significantly attenuated EBOV-driven luciferase expression in the primary ISG screen, we fitted a linear model for a balanced design to the data (function *aov*), comparing the ISGs and the negative control. Next, we used the package multcomp to build a contrast matrix comparing each ISG to the control, and compared them using Dunnett's test with a one-sided alternative hypothesis (i.e., that overexpression of the ISG resulted in a lower signal than the control; function glht). Finally, we adjusted the *p*-values to account for family-wise errors using Holm's method. Adjusted *p*-values smaller than 0.05 were considered statistically significant. For all other data, the Student's two-tailed, paired, and unpaired *t*-test was used to assess statistical differences between samples. Significance levels were set at $p \leq 0.05$. Unless otherwise stated, all experiments were performed independently at least three times with similar results obtained each time.

**Reporting summary**. Further information on research design is available in the Nature Research Reporting Summary linked to this article.

## Data availability

The source data underlying Figs. 1–3, 4 (except 4b), 5 (except 5a), and 6 (except 6c), and Supplementary Figs. 1–11, 12c, 13–18, and 20 are provided as a Source Data file. All of the other data, such as immunofluorescence images supporting the findings of this study, are available within the article, supplementary information files, and from the corresponding authors upon reasonable request. Information on the prediction of *CCDC92* gene expression in human tissues is available from the GeneCards database and the Human Protein Atlas database. Information on the predicted coiled-coil domains of CCDC92 is available from Jpred4. Source data are provided with this paper.

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

## Acknowledgements

We thank Susan Watson for editing along with John Kunert for technical support. This work was supported in part by grants AI081680, UAI106772, AI091707, and DK082155 from the National Institute of Allergy and Infectious Diseases, National Institutes of Health.

## Author contributions

M.K., L.H.-B., P.J.H., and Y.K. designed the experiments. M.K., L.H.-B., and P.J.H. performed the experiments. M.O. established the HEK-293T VP30/L cell line. J.W.S. and C.M.R. provided the ISG library and edited the manuscript. M.K., L.H.-B., P.J.H., T.J.S.L., and Y.K. analyzed the data. M.K., L.H.-B., P.J.H., G.N., and Y.K. wrote the manuscript.

## Competing interests

The authors declare no competing interests.
