## [Peer Review File · Nature Communications]

Reviewers' Comments:

Reviewer #1:

Remarks to the Author:

Review of Kuroda et al.

The revised study by Kuroda et al. defines a number of interferon stimulated genes (ISGs) that inhibit replication of a VP30-deletion Ebola virus (EBOV). Additional studies then clarify at what replication steps a subset of ISGs act. Finally, detailed analysis of one ISG, CCDC92, is presented. The conclusion is that CCDC92 interacts with EBOV nucleoprotein (NP), thereby disrupting NP incorporation into virus particles. Overall, the study is of general interest and the conclusions, if definitive, would be significant. However, several aspects of the study, particularly those concerning CCDC92 mechanism of action, need additional attention. Specific concerns are highlighted below.

Specific Comments:

1. Line 106. Please clarify what specifically is meant by "nonspecific...effects" of ISG expression.
2. Line 156 and Figure 2F. The text states that BTN3A3 expression inhibits viral titer by 50%. However, the graph y-axis is labeled as Log10. If this is a log scale, it appears that titers are reduced ~ 100 -fold. Is this a linear scale or log scale? A 100-fold drop is obviously significant. A 50% drop may not be biologically meaningful.
3. Figure 3. It is not clear that the entry assay used in Figure 3 actually measures viral entry. The readout of the assay is luciferase expressed from a VSV backbone. Inhibition of VSV transcription or replication would score positive in the assay used. It would be helpful to also test the ISGs against GP-pseudotyped lentiviruses, or better yet, Ebola virus VLPs containing a beta-lactamase tagged VP40. There are established assays to measure entry into the cell of beta-lactamase.
4. Many of the assays in the manuscript use luciferase as a readout. The ISGs should therefore be tested for inhibition of luciferase activity.
5. Because RNA and protein synthesis are coupled, the experiments in Figure 4a

and 4b do not distinguish between a block in viral transcription, genome replication and protein synthesis. Time course studies and examination of RNA synthesis in the presence of cycloheximide could clarify these points.

6. The impact of CCDC92 on NP-VP40 interaction is very modest. Inhibition of NP incorporation into VLPs is much more impressive. Therefore, the claim that disrupted NP-VP40 interaction accounts for the inhibition of NP incorporation into VLPs is not convincing. Further complicating interpretation of this experiment, a significant amount of VP40 precipitates in the absence of NP.

7. Figure 5C. The NP deletion mutants used likely cut the protein in the middle of important structural domains. A binding site might be present within a particular construct but lose interaction with CCDC92 due to improper folding. Several NP structures are now available. The mapping should be repeated using mutants based on distinct, separately folding domains of NP.

8. Figure 5C. For NP-CCDC92 interaction studies, only loss of binding data is presented. The authors should try to define a subdomain of NP that is sufficient to bind CCDC92.

9. Figure 6d does not definitively demonstrate formation of a CCDC92-NP-VP35 complex. CCDC92 could be interacting independently with VP35 and with NP. Further controls, such as CCDC92+VP35 coIP, could further resolve this issue.

Reviewer #2:

Remarks to the Author:

In this work the authors identify novel ISGs that restrict EBOV infection. The authors successfully identify multiple relevant genes, confirm their results, and superficially describe the mechanism by which overexpression of one these genes restrict infection. The authors are successful in addressing most of the reviewer comments, but there are some concerns on the new mechanistic studies.

1.) The authors claim that CCDC92 disrupts the interaction between VP40 and NP is based on combination of cellular redistribution of NP in the presence of CCDC92 and co-IP data. Based on the immunofluorescence it appears that over expression of CCDC92 causes a reduction in VP40, this could also explain the decrease in VP40 CO-IP with NP.

2.) Another question raised by the immunofluorescence data is the apparent change in cellular localization of CCDC92 from Figure 4 where it appears to be

localized along the cellular membrane, compared to Figures 5 and 6 where it appears to be cytosolic.

3.) Using deletion constructs, the authors conclude that AA 2-150 of NP are necessary for CCDC92 binding. In this study an AA 2-150 segment of NP should be assayed to demonstrate that this domain is sufficient for binding. In addition, the focus of this work is on the ISGs, so there should be comparable studies done with CCDC92 to define its interacting domain.

4.) To show that CCDC92 sequesters viral complexes to inclusion bodies, the authors should use immunofluorescence to identify the cellular localization of CCDC92, NP, VP35, VP30 during an infection, rather than overexpression. At the very least since the authors hypothesize that the CCDC92/NP interaction is what drives this this experiment should be repeated in the absence of NP or with binding mutants.

We would like to thank the reviewers for their insightful comments and suggestions. Our point-by-point responses are detailed below:

Reviewer #1

1. Line 106. Please clarify what specifically is meant by "nonspecific...effects" of ISG expression.

Response: We changed this sentence to simply state "exhibited cytotoxic effects", which was our intended meaning (line 94).

2. Line 156 and Figure 2F. The text states that *BTN3A3* expression inhibits viral titer by 50%. However, the graph y-axis is labeled as Log_{10} . If this is a log scale, it appears that titers are reduced ~100-fold. Is this a linear scale or log scale? A 100-fold drop is obviously significant. A 50% drop may not be biologically meaningful.

Response: This was an error on the y-axis of Figure 2F. It is now corrected to state " $\times 10^5$ ffu/ml". Therefore, the virus titer was reduced by 50% ($p = 0.024$). The p value has been added to the manuscript (line 157).

3. Figure 3. It is not clear that the entry assay used in Figure 3 actually measures viral entry. The readout of the assay is luciferase expressed from a VSV backbone. Inhibition of VSV transcription or replication would score positive in the assay used. It would be helpful to also test the ISGs against GP-pseudotyped lentiviruses, or better yet, Ebola virus VLPs containing a beta-lactamase tagged VP40. There are established assays to measure entry into the cell of beta-lactamase.

Response: We performed additional experiments to examine the inhibition of virus entry by AKT3, FLT1, and ODC1 and to ensure that the inhibition observed was not due to the inhibition of the reporter gene in the assay.

With the pseudotyped VSV system, AKT3 and FLT1 did not inhibit entry mediated by the glycoproteins of Lassa virus and influenza virus, demonstrating these two ISGs are specific entry inhibitors of VSV and Ebola and that the inhibition did not correlate with VSV-driven luciferase expression (lines 178–182 and Supplementary Figure 7a–c).

ODC1 inhibited entry mediated by the glycoproteins of all viruses (VSV, Ebola, Lassa, and influenza) tested in the pseudotyped VSV system; therefore, we verified the antiviral activity with a retrovirus pseudotyped with the VSV glycoprotein (G) and that expresses GFP. Co-expression of ODC1 and the retrovirus virus vector that expresses GFP (pMXs-IRES-GFP) did not reduce GFP expression in transfected cells when compared to cells transfected with pMXs-IRES-GFP and an empty control vector. However, in cells expressing ODC1 that were transduced with the VSV-G retrovirus-GFP, we did observe a reduction in GFP expression when compared to cells transfected with an empty control vector and transduced with the VSV-G retrovirus-GFP (lines 183–192 and Supplementary Figure 8a–b). From these new data, we concluded that the inhibitory effect of ODC1 is specific to virus entry and not to the expression of the reporter gene.

4. Many of the assays in the manuscript use luciferase as a readout. The ISGs should therefore be tested for inhibition of luciferase activity.

Response: We have added new data in Supplementary Figures 1a–c on the effects of individual ISGs on vector-based expression of firefly luciferase and Renilla luciferase. We detected no significant reduction in the expression of either luciferase reporter gene by any of the ISGs relative to the expression of a control gene like GFP (lines 97–98).

5. Because RNA and protein synthesis are coupled, the experiments in Figure 4a and 4b do not distinguish between a block in viral transcription, genome replication and protein synthesis. Time course studies and examination of RNA synthesis in the presence of cycloheximide could clarify these points.

Response: As suggested by the reviewer, we repeated the NP mRNA transcription experiment in cells expressing CCDC92 and treated with cycloheximide (Supplementary Figure 15a–c). From these new data, we concluded that CCDC92 inhibits virus transcription (lines 305–314).

6. The impact of CCDC92 on NP-VP40 interaction is very modest. Inhibition of NP incorporation into VLPs is much more impressive. Therefore, the claim that disrupted NP-VP40 interaction accounts for the inhibition of NP incorporation into VLPs is not convincing. Further complicating interpretation of this experiment, a significant amount of VP40 precipitates in the absence of NP.

Response: We repeated the co-immunoprecipitation experiment between NP and VP40 with and without CCDC92 a total of three times. A representative blot is shown in Figure 5b and a graph of the averaged relative

band intensities is shown in Supplementary Figure 14a. Given this new data set, we feel confident that CCDC92 significantly inhibits the interaction between NP and VP40.

7. *Figure 5C. The NP deletion mutants used likely cut the protein in the middle of important structural domains. A binding site might be present within a particular construct but lose interaction with CCDC92 due to improper folding. Several NP structures are now available. The mapping should be repeated using mutants based on distinct, separately folding domains of NP.*

Response: As suggested by the reviewer, we generated two NP deletion mutants based on published structural regions of NP that are important for NP oligomerization and RNA encapsidation: the N-terminal lobe of NP (amino acids 39–240) and the C-terminal lobe of NP (amino acids 241–405). Using these mutants, we repeated the immunoprecipitation assay with CCDC92 and found that the N-terminal lobe of NP was sufficient to bind to CCDC92 (Figure 4c and lines 253–259).

8. *Figure 5C. For NP-CCDC92 interaction studies, only loss of binding data is presented. The authors should try to define a subdomain of NP that is sufficient to bind CCDC92.*

Response: Unfortunately, we could not further define the binding subdomain of NP that interacts with CCDC92; this may be due to improper folding of the NP amino acids 39–240.

To further examine the interaction between NP and CCDC92, we did generate two CCDC92 deletion constructs based on the coiled-coil domains within the protein, and defined the second coiled-coil domain (amino acids 81–151) as the interaction domain for its interaction with NP (Figure 4d, lines 260–269). These CCDC92 deletion mutants were also used in various other assays described in the manuscript (Figures 4e, 5b, and 5e; Supplementary Figures 12d–e, 14a–b, and 19a–b; lines 270–277, 287–295, 315–320, and 353–358).

9. *Figure 6d does not definitively demonstrate formation of a CCDC92-NP-VP35 complex. CCDC92 could be interacting independently with VP35 and with NP. Further controls, such as CCDC92+VP35 coIP, could further resolve this issue.*

Response: In the original Figure 6d, we had control cells that expressed CCDC92 (FLAG-tagged) and VP35 only and we were unable to detect any interaction between CCDC92 and VP35 without NP; these data demonstrate that there is no independent interaction between CCDC92 and VP35.

Reviewer #2

1.) *The authors claim that CCDC92 disrupts the interaction between VP40 and NP is based on combination of cellular redistribution of NP in the presence of CCDC92 and co-IP data. Based on the immunofluorescence it appears that over expression of CCDC92 causes a reduction in VP40, this could also explain the decrease in VP40 CO-IP with NP.*

Response: We are uncomfortable stating that the overexpression of CCDC92 causes a reduction in VP40 expression based on the immunofluorescence data because they are not quantitative data. In addition, the western blot data in cells co-transfected with NP and CCDC92 do not show a reduction in cellular VP40 expression when CCDC92 is expressed (Figure 5b and Supplementary Figure 14b).

2.) *Another question raised by the immunofluorescence data is the apparent change in cellular localization of CCDC92 from Figure 4 where it appears to be localized along the cellular membrane, compared to Figures 5 and 6 where it appears to be cytosolic.*

Response: The general conclusion we made in the manuscript is that CCDC92 and NP co-localize in cells. We have now added additional images showing the following: 1) CCDC92 is localized in the cytoplasm of Huh7.0 cells in the absence of expressed viral proteins (i.e., no virus infection; Figure 4b, right most panel), and 2) NP and CCDC92 co-localize in viral inclusion bodies during infection (Figure 4b, left panels). We acknowledge that the localization of CCDC92 in HEK-293T cells is more along the cellular membrane (Figure 5a), but this is when NP and VP40 are expressed from vectors (statement added; line 282).

3.) *Using deletion constructs, the authors conclude that AA 2-150 of NP are necessary for CCDC92 binding. In this study an AA 2-150 segment of NP should be assayed to demonstrate that this domain is sufficient for binding. In addition, the focus of this work is on the ISGs, so there should be comparable studies done with CCDC92 to define its interacting domain.*

Response: As suggested by Reviewer #1, we repeated the studies to define the domain of NP that is required for the interaction with CCDC92 by using NP deletion mutants based on the NP structure. We found that amino

acids 39–240 of NP are important for its interaction with CCDC92 (Figure 4c). We unfortunately could not show binding of NP amino acids 39–240 with CCDC92 due to the low expression of NP amino acids 39–240, which may be caused by improper folding and resulting protein instability.

Nevertheless, as recommended by this reviewer, we were able to generate deletion constructs for CCDC92. Two coiled-coil domains (amino acids 20–72 and 81–151) are predicted in CCDC92 by the online prediction software Jpred4. Therefore, we generated two FLAG-tagged CCDC92 deletion mutants, each lacking one of these domains (FLAG-CCDC92 Δ 20-72 and FLAG-CCDC92 Δ 81-151), and repeated our assays with wild-type CCDC92 and these deletion mutants (see Figures 4e, 5b, and 5e; Supplementary Figures 12d–e, 14a–b, and 19a–b; lines 270–277, 287–295, 315–320, and 353–358). We found that the second coiled-coil domain (amino acids 81–151) of CCDC92 is responsible for its interaction with NP. This deletion mutant lost its antiviral activity against Ebola virus, no longer co-localized with NP, and did not inhibit virus transcription in the minireplicon assay (Figures 4e, 5b, and 5e; Supplementary Figures 12d–e, 14a–b, and 19a–b; lines 270–277, 287–295, 315–320, and 353–358).

4.) To show that CCDC92 sequesters viral complexes to inclusion bodies, the authors should use immunofluorescence to identify the cellular localization of CCDC92, NP, VP35, VP30 during an infection, rather than overexpression. At the very least since the authors hypothesize that the CCDC92/NP interaction is what drives this this experiment should be repeated in the absence of NP or with binding mutants.

Response: We have performed immunofluorescence experiments with cells expressing wild-type CCDC92 or CCDC92 deletion mutants with and without virus infection. As demonstrated in Figure 4b and described in lines 247–252, NP and CCDC92 co-localize in NP inclusion bodies in infected cells, whereas the localization of CCDC92 in non-infected cells is diffuse in the cytoplasm.

A similar localization pattern was observed with NP and CCDC92 Δ 20-72 in infected cells (Supplementary Figure 12d); the CCDC92 deletion mutant that still interacts with NP in co-immunoprecipitation experiments (Figure 4d). However, the other CCDC92 deletion mutant (Δ 81-151) that does not interact with NP (Figure 4d), did not co-localize with NP in infected cells (Supplementary Figure 12e).

Similar results with the CCDC92 deletion mutants were observed in cells expressing the viral proteins (NP, VP35, VP30, and L) that are required to form the viral inclusion bodies (lines 353–358; Supplementary Figure 19a–b). Collectively, these data along with our other data support the hypothesis that the interaction of NP and CCDC92 correlates with the antiviral activity of CCDC92.

Reviewers' Comments:

Reviewer #1:

Remarks to the Author:

The revised study by Kuroda et al. defines a number of interferon stimulated genes (ISGs) that inhibit replication of a VP30-deletion Ebola virus (EBOV). Additional studies then clarify at what replication steps a subset of ISGs act. Finally, detailed analysis of one ISG, CCDC92, is presented. The conclusion is that CCDC92 interacts with EBOV nucleoprotein (NP), impairing viral transcription and virion production. The study is now much more thorough than in its prior iteration. The conclusions are now well-supported by the data.

Reviewer's Comments

Reviewer #1 (Remarks to the Author): The revised study by Kuroda et al. defines a number of interferonstimulated genes (ISGs) that inhibit replication of a VP30-deletion Ebola virus (EBOV). Additional studies then clarify at what replication steps a subset of ISGs act. Finally, detailed analysis of one ISG, CCDC92, is presented. The conclusion is that CCDC92 interacts with EBOV nucleoprotein (NP), impairing viral transcription and virion production. The study is now much more thorough than in its prior iteration. The conclusions are now wellsupported by the data.

Response: We appreciate the reviewer's comments and insights to help improve our manuscript.